# The Application of Carotenoid-Coated Chitosan Nanoparticles to Reduce the PAHs Stress on Spinach Growth

**Jin Zhang** [1,2,†]**, Menghan Cui** [1,2,†]**, Ran Tao** [1,2]**, Yifan Yao** [1]**, Jiangang Han** [1,2,*] **and Yu Shen** [1,2]

1   Co-Innovation Center for Sustainable Forestry in Southern China, College of Biology and the Environment, Nanjing Forestry University, Nanjing 210037, China; 18262397885@163.com (J.Z.); mhancui@163.com (M.C.); nature19980626@gmail.com (R.T.); ivanyyfan@163.com (Y.Y.); sheyttmax@hotmail.com (Y.S.)
2   National Positioning Observation Station of Hung-tse Lake Wetland Ecosystem in Jiangsu Province, Nanjing 210037, China
*   Correspondence: jianganghan310@outlook.com
†   These authors contributed equally to this work.

**Abstract:** Polycyclic aromatic hydrocarbons (PAHs) pose risks to human and animal health, and their accumulation in crops is a concern for the food chain in the environment. Nanoparticles (NPs) have shown potential for chemical delivery and can be used to enhance plant resistance to PAHs. In this study, carotenoid-coated chitosan nanoparticles (CCNPs) loaded with $\beta$-carotene were prepared and applied to spinach grown in PAH-contaminated soil. The size of the CCNPs varied based on reaction conditions with temperature, TPP, and pH, with sizes ranging from 260 to 682 nm. After four weeks of treatment, the spinach showed varying growth responses depending on the specific CCNP treatment. The treatment with CCNPs prepared at 20 °C, pH 6, and 10 mg/mL TPP resulted in the best spinach growth, while the treatment at 40 °C, pH 6, and a TPP concentration of 20 mg/mL hindered growth; and the growth ration increased by over 47.4% compared to the normal growing spinach, the final biomass reached 2.53 g per plant. In addition, phenanthrene (PHE) and pyrene (PYR) predominantly accumulated more in the spinach roots, with variations depending on the specific CCNP treatment. The exogenous application of CCNPs can reduce the PAH transfer to the shoots. The bioconcentration factors and transfer factors of PYR and PHE reduced differential movement within the spinach plants, and the spinach prefers PYR to PHE in biological accumulation. This study offers a new understanding of the mechanisms underlying NPs and PAHs interactions and NP's implications for crop protection and food safety.

**Keywords:** chitosan nanoparticles; coatings; carotenoid; PAHs; spinach





## 1. Introduction

Polycyclic aromatic hydrocarbons (PAHs) are a group of organic compounds composed of multiple fused aromatic rings [1]. They are formed during the incomplete combustion or pyrolysis of organic materials naturally in coal, crude oil, gasoline, etc. [2]. PAHs are ubiquitous in the environment and can be found in the air, water, soil, and various food sources, including agricultural products [3,4]. They are known to be persistent in the environment, bioaccumulate in organisms, and pose potential health risks to both humans and animals [5]. Thus, it is important to reduce the bioaccumulation of PAHs in crops.

PAHs can have detrimental effects on agricultural ecosystems [6]; PAHs can be absorbed by plant roots from contaminated soil or water and accumulate in plant tissues [7,8], leading to a decrease in the biomasses of both roots (21.0%–42.7%) and leaves (6.4%–22.1%) grown in PAH-contaminated soil [9]. If PAH-contaminated soil is present, plants growing in the soil can uptake PAHs through their root system [10]; it was found that the root can transfer PAHs through root pressure and transpiration from the root to the shoot [11], and the root xylem plays an important role in the PAHs transfer [12]. Moreover, it was reported that the PAHs accumulation in roots can reach 203 ng g$^{-1}$, which is two times

higher than that in the shoot tissue in crops [13]; and the cell wall fraction of the root is the major part of over 45% of PAHs adsorption at the subcellular level [14]. How to control and reduce the adsorption of PAHs from the soil is a breakthrough for crop protection in PAH-contaminated environments.

Nanoparticles (NPs) have emerged as valuable tools for chemical delivery due to their unique properties and potential applications [15,16]. The NPs size, from 1 to 100 nanometers, offers several advantages for delivering chemicals efficiently and effectively, and this technology holds the promise of the controlled release of agrochemicals and site-targeted delivery of various macromolecules needed for improved plant disease resistance, efficient nutrient utilization, and enhanced plant growth [17]. For example, a kind of poly(lactic-co-glycolic) acid NP was designed for the membrane of neural stem cell delivery in the ischemic brain and established a novel formulation of glyburide [18]; the 400.57 nm chitosan nanoparticles (CNPs) coated with hyaluronic-acid can be used for the delivery of dexamethasone for patients, and the final loaded ration can reach around 72.95% and 14.51% those are very useful for medicinal use [19]. NPs dual-coated with chitosan and albumin allowed sustained insulin release following their passage to simulated intestinal conditions [20]. As described, chitosan NPs present the potential to overcome the barriers to the oral delivery of protein drugs, leading to the development of platforms capable of improving their bioavailability [21].

Chitosan (chitin) is an environmentally friendly material that comes from the outer skeleton of *Crustacea* [22]. In addition, chitosan has been established as a non-toxic, biodegradable, and biocompatible compound, as recognized by the United States Food and Drug Association (US FDA) [23]. Furthermore, chitosan production offers an environmentally sustainable solution by utilizing bio-waste generated from the crustacean production industries. Globally, chitosan production amounts to approximately 6–8 million tons per year, with 1.5 million tons produced by Southeast Asian countries [24]. This approach contributes to a "zero-waste" food industry, benefiting both the economy and the environment [25]. By repurposing these by-products, chitosan serves as a valuable resource in various applications, including the synthesis of CCNPs, and underscores the potential of eco-friendly practices in fostering a more sustainable future. Ionic gelation is the most commonly used method for synthesizing CNPs, and this kind of polymeric nanoparticles has gained significant importance as they are biodegradable, biocompatible, and because formulation methods are more widely available with a large surface area-to-volume ratio [26]. It was found that chitosan-coated mesoporous silica nanoparticles at the seedling stage led to a 70% increase in the fruit yield of uninfected watermelon because of their high surface area [27]. A carrier system for paraquat using polymeric nanoparticles composed of chitosan/TPP can make paraquat less toxic and, therefore, allows safer control of weeds in agriculture due to its controlled release [28]. The increased surface area allows for greater interaction between the nanoparticles and the target chemicals, enabling efficient loading, encapsulation, and controlled release. Thus, it is possible to figure out a kind of chemical to increase PAHs resistance in plants from root absorption.

In a previous study, it was found that $\beta$-carotene is a kind of antioxidant performing a positive role in ROS scavenging when the plants were treated with PAHs [29], and the root application of $\beta$-carotene could significantly increase wheat's resistance to the PAHs [30]. Indeed, the direct application of $\beta$-carotene is challenging due to its sensitivity to heat, light, and oxidation [31]. In this study, we addressed this issue by designing CNPs capable of encapsulating $\beta$-carotene for use in PAH-contaminated soil. The small size of these nanoparticles not only minimizes systemic toxicity but also enhances their bioavailability, making them more effective in improving the PAH resistance of crops. By entrapping $\beta$-carotene within CNPs, we provide a protective shield, safeguarding it from degradation caused by environmental factors. This innovative approach offers a promising solution for utilizing $\beta$-carotene's benefits in enhancing crop resilience under PAH contamination, contributing to sustainable agricultural practices. We selected the most common vegetable, spinach (*Spinacia oleracea*), for the study, which is a leafy, green vegetable that originated

in Persia; it is considered healthy, as it is loaded with nutrients and antioxidants [32]. In addition, PAHs have stronger accumulation in leafy plants [33]. Therefore, with the best carrier and release performance of CNPs, $\beta$-carotene is expected to show its role in improving PAH resistance and protecting crop growth under a PAH-contaminated environment. Our study is important for PAH-contamination management in agricultural systems and minimizing the risk of plant uptake. Additionally, we study implementing good agricultural practices and avoiding the mitigation of PAH absorption by plants.

## 2. Material and Methods

### 2.1. Carotenoid-Coated Chitosan Nanoparticles (CCNPs) Preparation

The CNPs were prepared by ionotropic gelation with some modifications. First, 50 mL of a solution of chitosan (10 mg/mL; pH 4.0; 27 kDa; 75%–85% deacetylation), prepared in an aqueous solution of 1% acetic acid, was kept under vigorous stirring on a magnetic intelligent color display heating stirrer (TP-350$^+$; MIULAB Co., Ltd., Hangzhou, China). The reaction formula is presented below,

$$\text{C-NH}_3{}^+ - \text{Ac}^- + \text{Na}^+(\text{TPP} - \text{PO}^-) \rightarrow \text{C-NH}_3{}^+(\text{TPP} - \text{PO}^-) + \text{NaAC}$$

After the chitosan was dissolved totally, the tripolyphosphate (TPP) solution was added to the chitosan solution at the ratio of 1:5 ($w/w$), and the mixed solution was stirred for one hour at 1000 r/min to obtain chitosan nanoparticles (Figure 1a). In the preparation, two temperature levels (20 and 40 °C), two TPP concentrations (10 and 20 mg/mL), and two pH levels (5 and 6) were selected for the CCNP preparation (Table 1). Then the following, CCNPs were prepared with the CNPs and $\beta$-carotene, which were added to the CNP solution at a 1:2 ($w/w$) ratio with the ultrasonic of 40 KHz (KH-250E; Kunshan Hechuang Ultrasonic Instruments Co., Ltd., Shanghai, China) for 30 min. Then, the CCNP solution was filtered by the dialyzer (HCA000808; Union Carbide Co., Houston, TX, USA) at 10 kDa. Then, the CCNP solution was stored in the 4 °C fridge for the root ball application. The prepared CNPs and CCNPs were also characterized using transmission electron microscopy (TEM, JEM-2100 UHR, JEOL, Akishima, Japan). The NP solutions were dispersed properly with water, and a mixed nanoparticle solution was prepared by sonicating the solution for 10 min. We took a carbon-coated grid on a Whatman paper and added a 10 µL drop onto a grid using a micropipette for natural drying. The NP images are shown in Figure 1b,c.

### 2.2. Greenhouse Experiment

The spinach seeds used were commercial Changfeng seeds (Aishen Vegetable Seed Breeding Center, Qingxiang, China). The seeds were germinated in a nursery box at 25 °C and 70% humidity (RDN-1000D; Yanghui Equipment Co., Ltd., Ningbo, China). Every two seeds were placed in a plastic pot (9 cm × 7 cm × 6.5 cm) with four drainage holes of 1 cm diameter at the bottom, and the pots were placed in trays (51 cm × 28 cm × 6.5 cm) to prevent soil leakage. Two weeks later, when the seedlings had three true leaves, uniform seedlings were selected for the root ball application at Baima Greenhouse, Nanjing Forestry University, Nanjing, China.

PAH soil was prepared with phenanthrene (PHE) and pyrene (PYR) (Merck KGaA, Darmstadt, German); 500 ppm concentration of PHE and PYR were added into the peat soil (Pindstrup Mosebrug, Ryomgaard, Demark), and the final PAH concentration was 5 ppm. In the seedling transfer, the root ball application was treated with the CCNPs; in the process, the configured nano-colloid solution was ultrasonically shaken for 30 min, and 5 mL of 8 kinds of CCNPs solution was injected into the root ball; the root ball was completely infiltrated with CCNPs. The seedlings were subsequently placed on the skeletonized shelf until there was no dripping liquid and then transplanted to the pots (11 cm × 10 cm × 13 cm); during the transplanting, the bulb was completely embedded in the soil and mulched to complete the inter-root exposure of CCNPs. Each treatment has three biological replicates.

After four weeks, the treated spinach was harvested, and the phenotype was recorded at the same time; afterward, the spinach was carefully separated from the soil and gently rinsed four times with tap water, followed by deionized water to remove soil adhering to the plant surface. After drying, the shoots and roots of spinach were separated with scissors and weighed. Three replicates of each treatment were applied, and a nutrient solution was applied once in the second week after treatment.

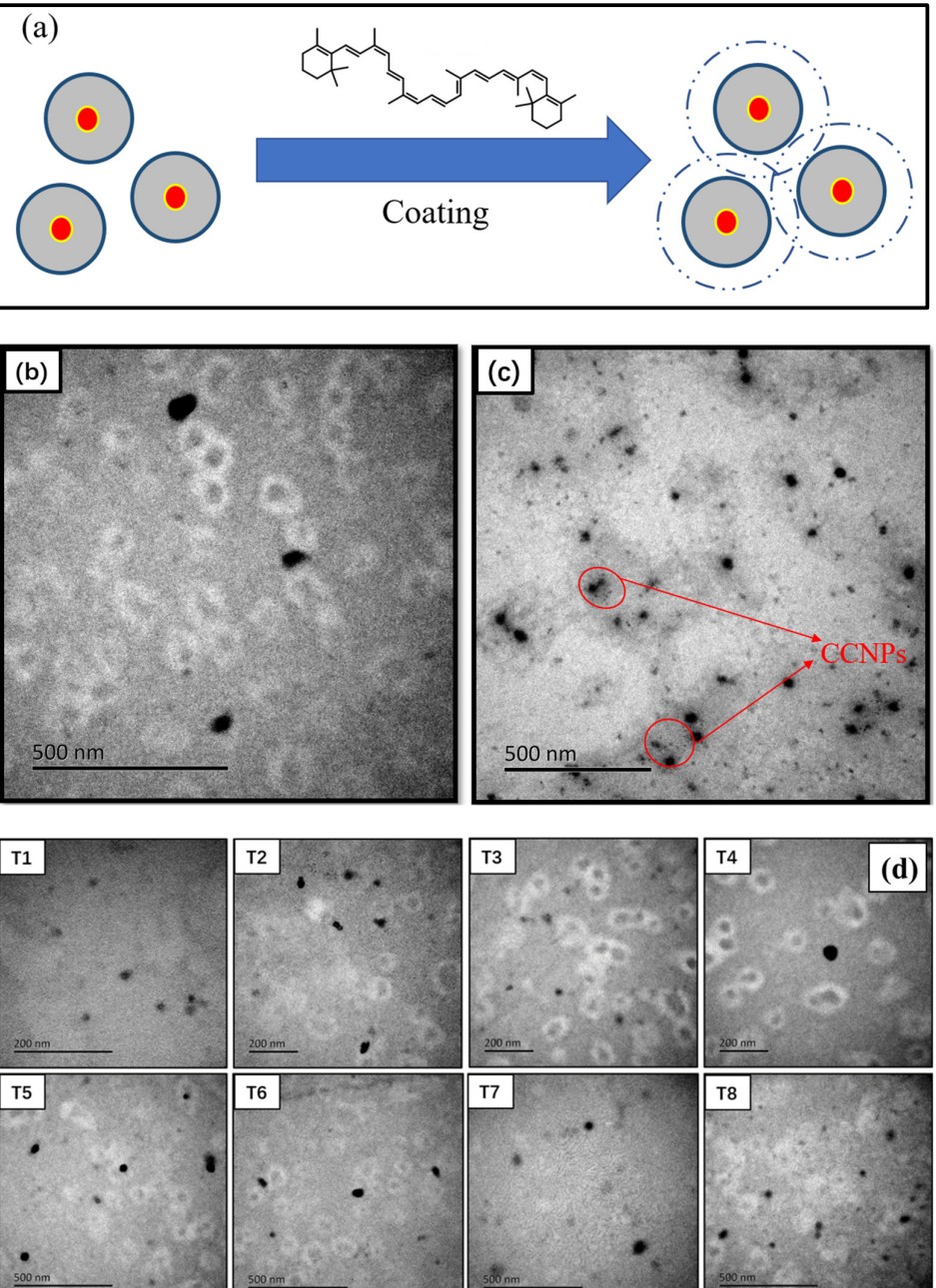

**Figure 1.** Mechanism process of carotenoid-coated chitosan nanoparticles (CCNPs) (**a**); and the representative TEM images of bare chitosan nanoparticles (CNPs) (**b**), and CNPs generated by ionic gelation at a reaction temperature of 20 °C, a TPP solution concentration of 10 mg/mL, and a pH of 6 (i.e., T2) coated with $\beta$-carotene (CCNPs) (**c**); and TEM analysis of CNPs cross-linked by ionotropic gelation with TPP. (Note: The treatments of T1–T8 are shown in Table 1). (**d**) Note, T1, 20 °C, 10 mg/mL TPP, pH = 5; T2, 20 °C, 10 mg/mL TPP, pH = 6; T3, 20 °C, 20 mg/mL TPP, pH = 5; T4, 20 °C, 20 mg/mL TPP, pH = 6; T5, 40 °C, 10 mg/mL TPP, pH = 5; T6, 40 °C, 10 mg/mL TPP, pH = 6; T7, 40 °C, 20 mg/mL TPP, pH = 5; and T8, 40 °C, 20 mg/mL TPP, pH = 6.

**Table 1.** Treatment information for comparison experiment.

| Treatments | T/°C | TPP (mg/mL) | pH |
|---|---|---|---|
| T1 | 20 | 10 | 5 |
| T2 | 20 | 10 | 6 |
| T3 | 20 | 20 | 5 |
| T4 | 20 | 20 | 6 |
| T5 | 40 | 10 | 5 |
| T6 | 40 | 10 | 6 |
| T7 | 40 | 20 | 5 |
| T8 | 40 | 20 | 6 |

*2.3. PAHs Extraction and Analysis*

The extraction and purification procedure for PAHs was based on that reported by Gao et al. [2]. Specifically, the chopped plant samples were extracted with an extraction agent (acetone: dichloromethane = 2:1, *v/v*) using an ultrasonic water bath for 30 min, and this step was repeated three times. The combined extracts were passed through a silica gel column (silica gel 3 g, anhydrous sodium sulfate 3 g) and eluted with 10 mL of 1:1 (*v/v*) dichloromethane and n-hexane. The filtrate was passed through a rotary evaporator (RE-25A; Yarong Biochemical Instrument Factory, Shanghai, China), exchanged with 2 mL of methanol, filtered through a 0.45 μm Teflon membrane, and transferred into a 2 mL sample vial. PAH concentrations were measured by HPLC (UltiMate 3000 HPLC; Thermo Co., Ltd., Waltham, MA, USA). The HPLC conditions were as follows: the pump model was LPG-3400 SDN, the UV detector model was VWD-3100, and the column was a 4.6 mm × 150 mm C18 column with a temperature of 30 °C. The mobile phase was methanol/water (80/20, *v/v*) at a flow rate of 1.0 mL/min. The injection volumes for phenanthrene and pyrene were set to 10 and 40, respectively, and the UV detection wavelength of phenanthrene was 254 nm, and that of pyrene was 234 nm. The peak areas were quantified by the external standard method.

*2.4. Bioconcentration Factor (BCF) and Translocation Factor (TF)*

The ability of CCNPs to reduce PAHs accumulation is by measuring the bioconcentration factor (BCF) (1) and translocation factor (TF) (2), defined as the ratio of PAH concentration in plant shoots to roots and the ratio of PAH concentration in plant roots to soils, respectively [33]; and the results are presented in Table 2.

$$BCF = \frac{\text{PAHs concentration in plants}}{\text{PAHs concentration in sediment}} \tag{1}$$

and

$$TF = \frac{\text{PAHs concentration in shoot}}{\text{PAHs concentration in root}} \tag{2}$$

**Table 2.** Bioconcentration factors (BCFs) and transfer factor (TFs) of PAHs in the shoots and roots of spinach.

| Treatment | Bioconcentration Factor (BCF) | | Transfer Factor (TF) | |
|---|---|---|---|---|
| | $BCF_{PYR}$ | $BCF_{PHE}$ | $TF_{PYR}$ | $TF_{PHE}$ |
| T1 | 0.390 | 0.198 | 0.011 | 0.063 |
| T2 | 0.088 | 0.035 | 0.007 | 0.079 |
| T3 | 0.329 | 0.380 | 0.166 | 0.040 |
| T4 | 0.180 | 0.165 | 0.085 | 0.026 |
| T5 | 0.188 | 0.158 | 0.032 | 0.034 |
| T6 | 0.567 | 0.084 | 0.009 | 0.064 |
| T7 | 0.200 | 0.191 | 0.268 | 0.061 |
| T8 | 0.226 | 0.037 | 0.049 | 0.037 |

### 2.5. Statistics

Statistical analysis was performed using the Excel (Microsoft, Redmond, WA, USA) (version 2019), IBM SPSS Statistics 25.0 (IBM, Armonk, NY, USA). Sampling and chemical analyses were examined in triplicate to decrease the experimental errors and to increase the experimental reproducibility. The confidence of the data generated in the present investigations was analyzed by standard statistical methods to determine the mean values and standard deviation (S.D.). Descriptive Statistics were applied to assess the normality of the distribution, and the test data meets the normality. The differences among the treatments were analyzed by one-way ANOVA (LSD test).

## 3. Results

### 3.1. The CCNPs Characteristics

The sizes of CNPs were around 260–682 nm; the smallest CNPs occurred with a reaction temperature at 20 °C, a TPP concentration of 10 mg/mL, and the pH 6 (T2); and the largest were prepared with the reaction conditions of 40 °C, pH 6, and 10 mg/mL of TPP (Figure 1b,c). Based on the preparation of the coating NPs, β-carotene and ribonucleic acid (RNA) had similar functional groups as the hydrophobic effect, such as the nitrogenous bases, ribose, and phosphate groups of RNA and long carbon chain of β-carotene. Due to the similar chemical functions, it is considered that β-carotene can be coated on the CNPs like RNA (Figure 1a) [34], and CNPs are produced with β-carotene finally for the spinach growth experiment.

### 3.2. The Phenotype and Growth of the Spinach

After four weeks' treatment, we found that the spinach presented stress in the treatment of combined PAHs of PHE and PYR with no exogenous addition (CK), and the leaves went curly; the spinach presented growth limitations when treated with T3, T4, and T7 in the combined PAH contaminations, and those were shorter than that in the CK (Figure 2a). The spinach grew the least in the T7 treatment; the leaves were shriveled after the treatment with the CCNPs of different temperatures and pH, but the TPP concentration was focused at 20 mg/mL. The spinach grew the most in T1, T2, T5, and T8; the spinach growth was the best in T2, for which the CCNPs were prepared with a reaction temperature of 20 °C, 10 mg/mL TPP, and pH 6. The leaves in the PAHs treatment of PAH and PYR were larger and stronger when compared with the spinach treated with only PAHs (Figure 2a).

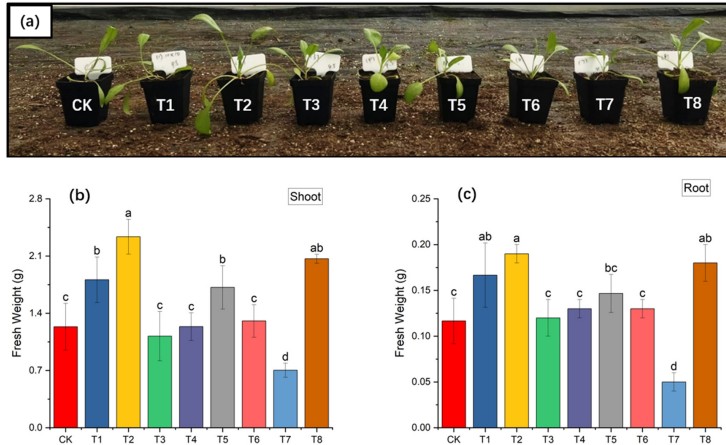

**Figure 2.** The phenotypes of spinach in the screening experiment of PAH-contaminated soil after 40 days (**a**), and the fresh weight of spinach shoots and roots after 40 days under CCNPs (**b,c**), respectively (Note, the error bars indicate the standard deviation ($n$ = 3); each sample with a different letter above indicates statistically significant differences with at least $p < 0.05$; CK was the blank control group, which grew naturally without reagent or contamination. The T1–T8 treatments are shown in Table 1).

It was recorded that the whole fresh weight of the spinach presented as the highest at 2.52 g in the treatment of 20 °C, pH 6, and the TPP concentration of 10 mg/mL (T2) with a shoot of 2.34 g and a root of 0.18 g; and the smallest, of 0.75 g, was recorded in the treatment of 40 °C, pH 5, and 20 mg/mL TPP (T7), with the shoot weighing 0.70 g and the root weighing 0.05 g (Figure 2b,c). In addition, the spinach's fresh weight of the shoot and root was 1.235 g and 0.11 g in the PAHs only treatment, respectively; and the shoot's fresh weights were 1.12, 1.24, and 0.70 g in the T3, T4, and T7 treatments, which were lower than that in the CK after 4 weeks. The shoot's fresh weight presented an increasing trend in the T2, T5, T6, and T8 treatments (Figure 2b). Despite the spinach root's fresh weight being lower than that in the CK of 0.06 g, the other root's fresh weights presented an increased response after the CCNPs were added; the fresh weight reached 0.12 to 0.19 g (seven kinds of CCNPs), respectively (Figure 2c).

### 3.3. The PAHs Concentration in Spinach

After 4 weeks of PAH treatment, the roots were the major location for PYR and PHE accumulation (Figure 3). In the treatment of CCNPs with a reaction temperature of 40 °C, 10 mg/mL TPP, and the solution with pH 6 (T6), the spinach root accumulated the highest ($p < 0.05$) PYR of 8.22 mg kg$^{-1}$, while the spinach root only accumulated 3.88, 0.97, 2.15, 1.30, 1.59, 1.29, and 2.08 mg kg$^{-1}$ when treated with the other CCNPs; the spinach root in T6 presented the highest at 8.22 mg kg$^{-1}$ among all treatments (Figure 3b). The PYR accumulated the lowest at 0.041 mg kg$^{-1}$ in treatment T2, and the PYR increased to the highest of 0.42 and 0.47 mg kg$^{-1}$ when treated with T3 and T7 after 4 weeks, respectively (Figure 3a). PHE accumulated more in the spinach shoots after 4 weeks of PAHs treatment, it reached 0.13 mg kg$^{-1}$, and the spinach shoot had similar contents of 0.10 and 0.11 mg kg$^{-1}$ when the spinach was treated with CCNPs of the 20 mg/mL TPP (T3 and T7), respectively; The PHE was lowest at 0.01 mg kg$^{-1}$ when the spinach was treated with CCNPs of 40 °C, pH 6, and the TPP concentration of 20 mg/mL (T8) (Figure 3c). PHE accumulated more in the roots when the spinach was treated with CCNPs of 20 °C, pH 5, and 20 mg/mL. For the TPP concentration of T3, when compared with the spinach root in the CK with 1.83 mg kg$^{-1}$, the PHE accumulated the least in the spinach root treated with the CCNPs of 20 °C, pH 6, and 10 mg/mL TPP (T2) and 40 °C, pH 6, and 20 mg/mL TPP (T8); the PHE concentrations are 0.32 and 0.36 mg kg$^{-1}$, respectively (Figure 3d).

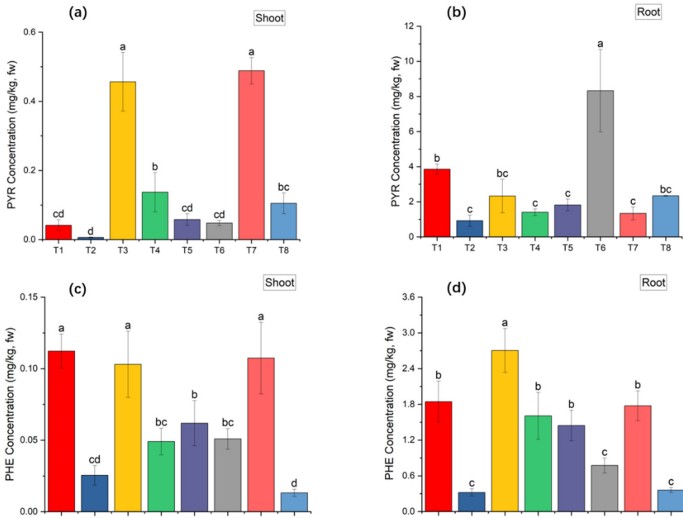

**Figure 3.** Concentrations of pyrene (PYR) and phenanthrene (PHE) in the shoots (**a**,**c**) and roots (**b**,**d**) of spinach (Note: the error bars indicate the standard deviation (*n* = 3); each sample with a different letter above indicates statistically significant differences with at least *p* < 0.05; CK was the blank control group, which grew naturally without reagent or contamination. The treatments T1–T8 are shown in Table 1).

### 3.4. BCFs and TFs

The BCFs and TFs of PYR and PAH are calculated in Table 2. It is shown that the PYR presented higher BCF in most treatments than PHE in the spinach, the $BCF_{PHE}$ is higher than $BCF_{PYR}$ only in T3, and $BCF_{PYR}$ is highest, at 0.567, in T6 and lowest, at 0.088, in T2. Meanwhile, the TF presented different trends than BCF in the spinach under PAH treatments. As shown in Table 2, the $TF_{PYR}$s are higher than the $TF_{PHE}$s with values of 0.166, 0.085, 0.268, and 0.049 in T3, T4, T7, and T8, respectively; and in other treatments, the $TF_{PYR}$s are lower than the $TF_{PHE}$s, with the values of 0.063, 0.079, 0.034, and 0.064 in T1, T2, T5, and T6, respectively.

## 4. Discussion

### 4.1. Preparation of CCNPs and PAHs Accumulation

The synthesis of CCNPs involved varying reaction conditions, resulting in particles of different sizes. The sizes of CNPs ranged from approximately 260 to 682 nm, with the smallest size observed at a reaction temperature of 20 °C, TPP concentration of 10 mg/mL, and pH 6. On the other hand, the largest size was obtained under the conditions of 40 °C, pH 6, and 10 mg/mL TPP (Figure 1). Among the CCNP treatments, the one prepared at 40 °C, 10 mg/mL TPP, and pH 6 (T6) showed the highest accumulation of PYR in spinach roots, with a concentration of 8.22 mg kg$^{-1}$, significantly higher ($p < 0.05$) than the control treatment without CCNPs (CK). The other CCNP treatments resulted in varying levels of PYR accumulation in spinach roots, ranging from 0.97 to 2.15 mg kg$^{-1}$, depending on the specific CCNP treatment.

In subsequent experiments, CCNPs were used to investigate the transport blocking of polycyclic aromatic hydrocarbons (PAHs), specifically PHE and PYR, in spinach plants. After four weeks of treatment, the concentration of PAHs in the spinach was assessed, revealing significant accumulation in the roots. Interestingly, the spinach shoots contained higher concentrations of PHE compared to the roots. The CCNP treatments at TPP concentrations of 20 mg/kg (T3 and T7) resulted in shoot PHE concentrations ranging from 0.10 to 0.11 mg kg$^{-1}$. The lowest shoot PHE accumulation of 0.01 mg kg$^{-1}$ was observed in the treatment with CCNPs prepared at 40 °C, pH 6, and 20 mg/mL TPP concentration (T8). In contrast, the accumulation of PAHs in the spinach roots varied depending on the specific CCNP treatment. For example, in the treatment with CCNPs prepared at 20 °C, TPP concentration of 5 mg/mL, and pH 6 (T3), the root PHE accumulation was higher compared to the CK. Conversely, the treatments with CCNPs prepared at 20 °C, pH 6, and 10 mg/mL TPP (T2), as well as 40 °C, pH 6, and 20 mg/mL TPP (T8), resulted in the lowest root PHE accumulation, with concentrations of 0.32 and 0.36 mg kg$^{-1}$, respectively. The data indicate that smaller CCNPs facilitated reduced PAH accumulation and transfer in spinach. Furthermore, no significant differences were observed in the concentration of PYR in shoots between samples T5 (0.058 mg kg$^{-1}$) and T6 (0.048 mg kg$^{-1}$), as well as in the concentration of PHE between samples T5 (0.052 mg kg$^{-1}$) and T6 (0.050 mg kg$^{-1}$). Similarly, there were no significant differences in the concentration of PYR in roots among samples T4 (1.3 mg kg$^{-1}$), T5 (1.59 mg kg$^{-1}$), and T7 (1.29 mg kg$^{-1}$), as well as in the concentration of PHE among samples T4 (1.61 mg kg$^{-1}$), T5 (1.53 mg kg$^{-1}$), and T7 (1.80 mg kg$^{-1}$). These findings suggest that reaction conditions may differentially influence root and shoot responses at various levels.

Commonly, CNPs are sensitive to temperature and pH [35]. Varying pH can affect the size of CNPs and probe the states of water in CNP hydrogels [36], while smaller CNPs have high water imbibing capability, minimal invasiveness, porous networks, and can mold perfectly into an irregular defect [37]. Temperature is another key factor for CNP synthesis; CNPs formed at high temperatures may have remaining associations, confirmed by their spontaneous recovery after breakup at low temperatures [35]. Moreover, chitosan treated at 25 °C possessed similar or weaker antibacterial activity compared to those at 4 °C, which can influence the CNP coating with $\beta$-carotene. In our study, we selected 20 °C as a reasonable temperature for CCNP synthesis and 20 °C with pH 6 as the ideal

condition for CCNP coating with $\beta$-carotene. TPP serves as a polymerization agent for NPs formation, and lower TPP concentrations were found to result in smaller CNP sizes. Our results match well with the previous report that the coated CNPs with nano-size can have biological effects on the crops via carrying materials to the target cells in plants [27]. With their nano-size, CCNPs exhibited increased adsorption capacity for PAHs. The preparation of CCNPs influenced the accumulation of PAHs in spinach plants, as specific conditions, such as reaction temperature, TPP concentration, and pH, played a role in determining the extent of PAH accumulation in the roots and shoots of the spinach plants. These findings highlight the importance of understanding the interaction between CCNPs and PAHs to assess their potential impact on plant health and food safety.

*4.2. CCNPs and PAHs Transfer*

The interaction between CCNPs and polycyclic aromatic hydrocarbons (PAHs) in the environment is crucial as it can influence the transfer of these contaminants within ecosystems. In this study, we investigated the transfer of CCNPs and PAHs, focusing on their movement within spinach plants. The synthesis of CCNPs resulted in particles of varying sizes, influenced by reaction conditions such as temperature, TPP, and pH [36]. The resulting CCNPs ranged in size from approximately 260 to 682 nm, with the smallest size observed under specific conditions (20 °C, 10 mg/mL TPP, and pH 6) and the largest size obtained under different conditions (40 °C, 10 mg/mL TPP, and pH 6) (Figure 1).

We then analyzed the transfer and accumulation of PAHs, specifically PHE and PYR, within the spinach plants. After four weeks of treatment, we observed significant PAH accumulation in the roots of the spinach plants. Notably, the treatment with CCNPs prepared at 40 °C, TPP concentration of 10 mg/mL, and pH 6 (T6) resulted in the highest accumulation of PYR in spinach roots, with a concentration of 8.22 mg kg$^{-1}$, significantly higher ($p < 0.05$) than the control treatment without CCNPs (CK). The other CCNP treatments led to varying levels of PYR accumulation in spinach roots, ranging from 0.97 to 2.15 mg kg$^{-1}$, depending on the specific CCNP treatment.

It was reported that coated CNPs have stronger adsorption potential than organic containments, such as paraquat, 4-nitrophenol, methyl orange, cango red, etc. [38]. A similar synthesized process was reported where the CNPs when coated with oxide metals and oligo, performed better at absorbing the PAHs in the environment [39,40]. Our results agree with the previous study; the CCNPs reduce the PAH transfer from soil to root. Regarding PHE accumulation, the spinach shoots exhibited higher concentrations of this PAH compared to the roots. After the four-week treatment period, the spinach shoots accumulated PHE concentrations ranging from 0.10 to 0.11 mg kg$^{-1}$ in the treatments with CCNPs prepared at a TPP concentration of 20 mg/mL (T3 and T7), respectively. The lowest shoot PHE accumulation of 0.01 mg kg$^{-1}$ was observed in the treatment with CCNPs prepared at 40 °C, pH 6, and 20 mg/mL TPP concentration (T8). In contrast, the accumulation of PAHs in the spinach roots varied depending on the specific CCNP treatment. For example, in the treatment with CCNPs prepared at 20 °C, TPP concentration of 5 mg/mL, and pH 6 (T3), the root PHE accumulation was higher compared to the control treatment (CK). Conversely, the treatments with CCNPs prepared at 20 °C, pH 6, and a TPP concentration of 10 mg/mL (T2), as well as 40 °C, pH 6, and 20 mg/mL TPP (T8), resulted in the lowest root PHE accumulation, with concentrations of 0.32 and 0.36 mg kg$^{-1}$, respectively.

The result indicates that the exogenous application of CCNPs can significantly reduce the movement of PAHs from the environment to the roots and shoots of spinach plants. CCNPs are known for their role in chemical transfer, and we found that their application increased resistance to PAH contamination. Understanding the interaction between CCNPs and PAHs is crucial for evaluating their potential impact on plant health and food safety.

## 5. Conclusions

These findings demonstrate that the reduction of PAHs transfer and accumulation within spinach plants by CCNPs is influenced by various factors, including the physico-chemical properties of the nanoparticles and the specific exposure conditions. Particularly, CCNPs exhibit better performance at room temperature and neutral pH, making them more suitable as carriers for $\beta$-carotene in plants compared to conditions of high temperature and acidic pH. Under appropriate reaction conditions, the synthesized CCNPs show significant protective effects on plants by efficiently providing and releasing $\beta$-carotene in PAH-contaminated environments. The decreased uptake and translocation of PAHs within plants can have significant implications for food safety and environmental health.

Furthermore, the main materials used in CCNP synthesis are derived from natural and biologically harmless sources, indicating minimal risk in consuming them. Additionally, their sustainability is enhanced by the abundance of the precursor materials used in their synthesis. However, further research is necessary to fully comprehend the mechanisms of transfer and potential risks associated with the interaction between CCNPs and PAHs in plant systems. This knowledge will aid in understanding the sustainable agricultural benefits of CCNPs and their potential applications in environmental remediation.

**Author Contributions:** Conceptualization, J.H. and Y.S.; Methodology, J.H.; Formal analysis, M.C.; Investigation, J.Z., M.C., R.T. and Y.Y.; Resources, J.H.; Data curation, J.Z.; Writing—original draft, M.C. and Y.S.; Writing—review & editing, Y.S. All authors have read and agreed to the published version of the manuscript.

**Funding:** The present work was carried out with the financial support of Agricultural Innovation Project (CX(22)3133) of Jiangsu Academy of Agricultural Sciences; Yu Shen thanks the Chinese Scholarship Council (CSC) supporting his study overseas.

**Institutional Review Board Statement:** Not applicable.

**Informed Consent Statement:** Not applicable.

**Data Availability Statement:** Not applicable.

**Conflicts of Interest:** The authors declare that they have no known competing financial interests or personal relationships that could have appeared to influence the work reported in this paper.

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
