# Peer review of "The Application of Carotenoid-Coated Chitosan Nanoparticles to Reduce the PAHs Stress on Spinach Growth"

_coatings, doi:10.3390/coatings13081404_

Round 1
Reviewer 1 Report
From the caption to figure 3 it can be seen that the vegetation experiments were carried out in triplicate. This should be reflected in the description of the research methods.
Each replication consisted of only two plants. How did the authors assess the normality of the distribution of all indicators (Fig. 2.3) to apply the classical statistical criteria (mean, standard deviation) with such a small sample? You need to justify the choice of methods.
How was the symmetry of the distribution of indicators revealed for the application of one-way analysis of variance?
On fig. Table 3 shows that there are no differences in the content of pyrene (PIR) in shoots in samples T5 and T6 and phenanthrene (PE) in samples T4, T5 and T6. In the roots in samples T4, T5 and T7 pyrene and in samples T1, T4 and T5 phenanthrene. What conclusions can be drawn from this?
Author Response
Reviewer #1
Q1. From the caption to figure 3 it can be seen that the vegetation experiments were carried out in triplicate. This should be reflected in the description of the research methods.
Answer: Thanks for your good suggestion to improve our manuscript quality. We added the sentence that “And each treatment has three biological replicates” in the revised manuscript on line 161 and 162.
Q2. Each replication consisted of only two plants. How did the authors assess the normality of the distribution of all indicators (Fig. 2.3) to apply the classical statistical criteria (mean, standard deviation) with such a small sample? You need to justify the choice of methods.
Answer: Thanks for your good comments. Given that each replication consisted of only two plants, the small sample size presents challenges in applying classical statistical criteria like mean and standard deviation, as well as assessing the normality of the distribution (Nakagawa and Cuthill, 2007). With such limited data points, traditional normality tests like the Shapiro-Wilk or Anderson-Darling tests may not be appropriate due to their low power for small samples. And we applied Descriptive Statistics to assess the normality of the distribution of all indicators. Although the sample size is small, it is still available to calculate basic descriptive statistics such as the mean and standard deviation. While these statistics might not provide a complete representation of the population, they can give some preliminary information about the data's central tendency and spread.
In our data, the results with the mean and standard deviation below, and all the mean and standard deviation belongs the range of maximum and minimum. And we added the description of the Descriptive Statistics to assess the normality of the distribution in the revised manuscript.
Descriptive Statistics of All the Data
|
|
mean |
SD |
maximum |
minimum |
|
Biomass Shoot |
1.50 |
0.52 |
2.58 |
0.64 |
|
Biomass Root |
0.13 |
0.05 |
0.21 |
0.04 |
|
PHE-Shoot |
0.07 |
0.04 |
0.14 |
0.01 |
|
PHE-Root |
1.31 |
0.84 |
3.06 |
0.27 |
|
PYR-Shoot |
0.16 |
0.12 |
0.59 |
0.005 |
|
PYR-Root |
2.92 |
2.39 |
10.91 |
1.02 |
Reference
Nakagawa S, Cuthill I C. Effect size, confidence interval and statistical significance: a practical guide for biologists[J]. Biological reviews, 2007, 82(4): 591-605.
Q3. How was the symmetry of the distribution of indicators revealed for the application of one-way analysis of variance?
Answer: Thanks for your good comment. One-way analysis of variance (ANOVA) assumes that the data in each group follows a normal distribution with equal variances, and symmetry is one of the indicators of normality (Boos and Brownie, 2004). And we applied Descriptive Statistics to assess the normality of the distribution.
Descriptive Statistics of All the Data
|
|
mean |
SD |
maximum |
minimum |
|
Biomass Shoot |
1.50 |
0.52 |
2.58 |
0.64 |
|
Biomass Root |
0.13 |
0.05 |
0.21 |
0.04 |
|
PHE-Shoot |
0.07 |
0.04 |
0.14 |
0.01 |
|
PHE-Root |
1.31 |
0.84 |
3.06 |
0.27 |
|
PYR-Shoot |
0.16 |
0.12 |
0.59 |
0.005 |
|
PYR-Root |
2.92 |
2.39 |
10.91 |
1.02 |
Reference
Boos D D, Brownie C. Comparing variances and other measures of dispersion[J]. Statist. Sci. 2004, 19 (4), 571 - 578.
Q4. On fig. Table 3 shows that there are no differences in the content of pyrene (PYR) in shoots in samples T5 and T6 and phenanthrene (PHE) in samples T4, T5 and T6. In the roots in samples T4, T5 and T7 pyrene and in samples T1, T4 and T5 phenanthrene. What conclusions can be drawn from this?
Answer: Thanks for your good suggestion to improve our manuscript quality. We analyzed the effects of multiple treatments on plant biomass or PAHs content, and one-way ANOVA was used to analyze differences between quantitative data and categorical data. And we added the description that “no significant differences were observed in the concentration of PYR in shoots between samples T5 (0.058 mg kg-1) and T6 (0.048 mg kg-1), as well as in the concentration of PHE between samples T5 (0.052 mg kg-1) and T6 (0.050 mg kg-1). Similarly, there were no significant differences in the concentration of PYR in roots among samples T4 (1.3 mg kg-1), T5 (1.59 mg kg-1), and T7 (1.29 mg kg-1), as well as in the concentration of PHE among samples T4 (1.61 mg kg-1), T5 (1.53 mg kg-1), and T7 (1.80 mg kg-1). These findings suggest that reaction conditions may differentially influence root and shoot responses at various levels.” In the revised manuscript from Line 296 to 303.

Reviewer 2 Report
The paper presents an interesting research work. In general, the paper is properly organized. However, some of the description needs to be explained in more detail. I think that the current paper could be accepted after major revision.
1. There are many grammatical mistakes in the manuscript. Sentences are not scientifically robust. Please revise the manuscript thoroughly.
2. What is the novelty of this article compared with other articles?
3. The authors should include the novelty of the work in terms of cost, reliability, and performance.
4. Keywords should be rewritten.
5. In the introduction section, It is strongly recommended to add a recent literature survey and the novelty of the present study. Furthermore, analysis techniques, wastewater technologies, and their applications for a sustainable environment should be cited. Research gaps should be highlighted more clearly, and future applications of this study should be added. The information provided needs to be well organized for a better understanding of the reader. Authors can get help from the following recent literature. Authors can get help from the following recent literature to improve the section:
doi.org/10.1007/s10924-022-02464-8
6. Please add future aspects of this study in view of current results.
7. Conclusions should be rewritten. The authors are only repeating the results.
Minor editing of English language required
Author Response
Reviewer #2
General Comments: The paper presents an interesting research work. In general, the paper is properly organized. However, some of the description needs to be explained in more detail. I think that the current paper could be accepted after major revision.
Answer: Thanks very much for your encouragement on our manuscript.
Q1. There are many grammatical mistakes in the manuscript. Sentences are not scientifically robust. Please revise the manuscript thoroughly.
Answer: Thanks for your good suggestion to improve our manuscript quality. We revised our manuscript carefully through sentences, hope our revision can meet your critical requirements.
Q2. What is the novelty of this article compared with other articles?
Answer: Thanks for your good comment. The novelty is very important for the research. And we highlighted the novelty in the introduction.
Indeed, the direct application of β-carotene is challenging due to its sensitivity to heat, light, and oxidation (Stutz et al., 2015). In this study, we addressed this issue by designing CNPs capable of encapsulating β-carotene for use in PAHs contaminated soil. The small size of these nanoparticles not only minimizes systemic toxicity but also enhances their bioavailability, making them more effective in improving the PAHs resistance of crops. By entrapping β-carotene within CNPs, we provide a protective shield, safeguarding it from degradation caused by environmental factors. This innovative approach offers a promising solution for utilizing β-carotene's benefits in enhancing crop resilience under PAHs contamination, contributing to sustainable agricultural practices.
Reference
Stutz H, Bresgen N, Eckl P M. Analytical tools for the analysis of β-carotene and its degradation products[J]. Free radical research, 2015, 49(5): 650-680.
Q3. The authors should include the novelty of the work in terms of cost, reliability, and performance.
Answer: We agreed with your comment of the novelty of the work in terms of cost, reliability, and performance, and we added the description in our revised manuscript.
In addition, Chitosan has been established as a non-toxic, biodegradable, and biocompatible compound, as recognized by the United States Food and Drug Association (US FDA) (Bernkop-Schnürch, et al., 2012). Furthermore, chitosan production offers an environmentally sustainable solution by utilizing bio-waste generated from the crustacean production industries. Globally, chitosan production amounts to approximately 6–8 million tons per year, with 1.5 million tons produced by Southeast Asian countries (Boutrif, 2004). This approach contributes to a "zero-waste" food industry, benefiting both the economy and the environment (Yan and Chen, 2015). By repurposing these by-products, chitosan serves as a valuable resource in various applications, including the synthesis of CCNPs, and underscores the potential of eco-friendly practices in fostering a more sustainable future.
Reference
Bernkop-Schnürch A, Dünnhaupt S. Chitosan-based drug delivery systems[J]. European journal of pharmaceutics and biopharmaceutics, 2012, 81(3): 463-469.
Boutrif E. Institutions Involved in Food Safety: Food and Agriculture Organization of the United Nations (FAO)[J]. Encyclopedia of Food Safety, 2014, 4(3077):354-358.
Yan N, Chen X. Don't waste seafood waste[J]. Nature, 2015, 524(7564):155-157.
Q4. Keywords should be rewritten.
Answer: We revised the key words that “Chitosan Nanoparticles, Coatings, Carotenoid, PAHs, Spinach.”
Q5. In the introduction section, It is strongly recommended to add a recent literature survey and the novelty of the present study. Furthermore, analysis techniques, wastewater technologies, and their applications for a sustainable environment should be cited. Research gaps should be highlighted more clearly, and future applications of this study should be added. The information provided needs to be well organized for a better understanding of the reader. Authors can get help from the following recent literature. Authors can get help from the following recent literature to improve the section:
doi.org/10.1007/s10924-022-02464-8
Answer: Thanks very much for your good instructions. We read the article “Insights into effective adsorption of lead ions from aqueous solutions by using chitosan-bentonite composite beads” carefully, and it inspired us on the analysis, applications, and the future of chitosan NPs. We revised the section in the manuscript.
Chitosan (chitin) is an environmentally friendly material which comes from the outer skeleton of Crustacea [22]; In addition, Chitosan has been established as a non-toxic, biodegradable, and biocompatible compound, as recognized by the United States Food and Drug Association (US FDA) [23]. Furthermore, chitosan production offers an environmentally sustainable solution by utilizing bio-waste generated from the crustacean production industries. Globally, chitosan production amounts to approximately 6–8 million tons per year, with 1.5 million tons produced by Southeast Asian countries [24]. This approach contributes to a "zero-waste" food industry, benefiting both the economy and the environment [25]. By repurposing these by-products, chitosan serves as a valuable resource in various applications, including the synthesis of CCNPs, and underscores the potential of eco-friendly practices in fostering a more sustainable future. And the ionic gelation is the most commonly used method for synthesizing CNPs, and this kind of polymeric nanoparticles have gained significant importance as they are biodegradable, biocompatible and because formulation methods are more widely available with large surface area-to-volume ratio [26]. It was found that chitosan-coated mesoporous silica nanoparticles at the seedling stage led to a 70% increase in the fruit yield of uninfected watermelon because of their high surface area [27]. And a carrier system for paraquat using polymeric nanoparticles composed of chitosan/TPP can make paraquat less toxic and used for safer control of weeds in agriculture due to its controlled release [28]. The increased surface area allows for greater interaction between the nanoparticles and the target chemicals, enabling efficient loading, encapsulation, and controlled release. Thus, it is possible to figure out a kind of chemicals for plant to increase the PAHs resistance from the root absorption.
In previous study, it was found that β-carotene is a kind of antioxidants performing positive role in ROS scavenging when the plants treated with PAHs [29], and the root application of β-carotene could significantly increase the wheat resistance to the PAHs [30]. Indeed, the direct application of β-carotene is challenging due to its sensitivity to heat, light, and oxidation [31]. In this study, we addressed this issue by designing CNPs capable of encapsulating β-carotene for use in PAHs contaminated soil. The small size of these nanoparticles not only minimizes systemic toxicity but also enhances their bioavailability, making them more effective in improving the PAHs resistance of crops. By entrapping β-carotene within CNPs, we provide a protective shield, safeguarding it from degradation caused by environmental factors. This innovative approach offers a promising solution for utilizing β-carotene's benefits in enhancing crop resilience under PAHs contamination, contributing to sustainable agricultural practices.
Reference
[22] Y. Faqir, J. Ma, Y. Chai, Chitosan in modern agriculture production, Plant, Soil and Environment, 67 (2021) 679-699.
[23] A. Bernkop-Schnürch, S. Dünnhaupt, Chitosan-based drug delivery systems, European journal of pharmaceutics and biopharmaceutics, 81(2012) 463-469.
[24] E. Boutrif, Institutions involved in food safety: Food and agriculture organization of the United Nations (FAO), Encyclopedia of Food Safety, 4(2014) 4354-358.
[25] N. Yan, X. Chen, Don' t waste seafood waste, Nature, 524(2015) 155-157.
[26] Z M. Åženol, S. ÅžimÅŸek, Insights into effective adsorption of lead ions from aqueous solutions by using chitosan-bentonite composite beads, Journal of Polymers and the Environment, 30(2022) 3677-3687.
[27] J. T. Buchman, W. H. Elmer, C. Ma, K. M. Landy, J. C. White, C. L. Haynes, Chitosan-coated mesoporous silica nanoparticle treatment of citrullus ianatus (Watermelon): Enhanced fungal disease suppression and modulated expression of stress-related genes, ACS Sustainable Chemistry & Engineering, 7(2019).
[28] R. Grillo, A. E. S. Pereira, C. S. Nishisaka, R. De Lima, K. Oehlke, R. Greiner, L. Fraceto, Chitosan/tripolyphosphate nanoparticles loaded with paraquat herbicide: An environmentally safer alternative for weed control, Journal of Hazardous Materials, 278(2014) 163171.
[29] Y. Shen, J. Li, R. Gu, L. Yue, H. Wang, X. Zhan, B. Xing, Carotenoid and superoxide dismutase are the most effective antioxidants participating in ROS scavenging in phenanthrene accumulated wheat leaf, Chemosphere, 197(2018) 513-525.
[30] Y. Shen, J. Li, S. Shi, R. Gu, X. Zhan, Application of carotenoid to alleviate the oxidative stress caused by phenanthrene in wheat, Environmental Science and Pollution Research, 26(2019) 3593-3602.
[31] H. Stutz, N. Bresgen, P M. Eckl, Analytical tools for the analysis of β-carotene and its degradation products, Free radical research, 49(2015) 650-680.
Q6. Please add future aspects of this study in view of current results.
Answer: Thanks for your good suggestion. We revised the future aspects of this study in the manuscript,
Furthermore, the main materials used in CCNPs synthesis are derived from natural and biologically harmless sources, indicating minimal risk in consuming them. Additionally, their sustainability is enhanced by the abundance of the precursor materials used in their synthesis. However, further research is necessary to fully comprehend the mechanisms of transfer and potential risks associated with the interaction between CCNPs and PAHs in plant systems. This knowledge will aid in understanding the sustainable agricultural benefits of CCNPs and their potential applications in environmental remediation.
Q7. Conclusions should be rewritten. The authors are only repeating the results.
Answer: Thanks very much for your good comment. We revised the conclusion in the manuscript.
These findings demonstrate that the reduction of PAHs transfer and accumulation within spinach plants by CCNPs is influenced by various factors, including the physicochemical properties of the nanoparticles and the specific exposure conditions. Particularly, CCNPs exhibit better performance at room temperature and neutral pH, making them more suitable as carriers for β-carotene in plants compared to conditions of high temperature and acidic pH. Under appropriate reaction conditions, the synthesized CCNPs show significant protective effects on plants by efficiently providing and releasing β-carotene in PAHs-contaminated environments. The decreased uptake and translocation of PAHs within plants can have significant implications for food safety and environmental health.
Furthermore, the main materials used in CCNPs synthesis are derived from natural and biologically harmless sources, indicating minimal risk in consuming them. Additionally, their sustainability is enhanced by the abundance of the precursor materials used in their synthesis. However, further research is necessary to fully comprehend the mechanisms of transfer and potential risks associated with the interaction between CCNPs and PAHs in plant systems. This knowledge will aid in understanding the sustainable agricultural benefits of CCNPs and their potential applications in environmental remediation.

Reviewer 3 Report
The presented article is focused on the application of carotenoid-coated chitosan nanoparticles to reduce the polycyclic aromatic hydrocarbon sstress on spinach growth. The performed work is interesting. English is fine. The structure of the manuscript is fine. However, before the publication it should be improved. For this purpose, please, see the comments below:
1. The abstract is too long. Indeed, it should present the most important information related to the research.
2. In the Introduction Section the Authors show the most important information on the nanoparticles. However, it does not include all relevant references. Indeed, more recently published papers should be cited.
3. The manuscript is prepared carelessly. For instance, the quality of the eqations 1 and 2 should be improved.
4. The obtained results should be compared with those presented in the literature.
5. The Conclusion Section id too short.
Author Response
Reviewer #3
General Comments: The presented article is focused on the application of carotenoid-coated chitosan nanoparticles to reduce the polycyclic aromatic hydrocarbon stress on spinach growth. The performed work is interesting. English is fine. The structure of the manuscript is fine. However, before the publication it should be improved. For this purpose, please, see the comments below:
Answer: Thanks very much for your encouragement on our manuscript.
Q1. The abstract is too long. Indeed, it should present the most important information related to the research.
Answer: Thanks for your good suggestion to improve our manuscript quality. We revised our abstract into 250 words.
ABSTRACT
Polycyclic aromatic hydrocarbons (PAHs) pose risks to human and animal health, and their accumulation in crops is a concern for the food chain from the environment. Nanoparticles (NPs) have shown potential for chemical delivery and can be used to enhance plant resistance to PAHs. In this study, carotenoid-coated chitosan nanoparticles (CCNPs) loaded with β-carotene were prepared and applied to spinach grown in PAH-contaminated soil. The size of the CCNPs varied based on reaction conditions with temperature, TPP, and pH, with sizes ranging from 260 to 682 nm. After four weeks of treatment, the spinach showed varying growth responses depending on the specific CCNP treatment. The treatment with CCNPs prepared at 20°C, pH 6, and 10 mg/mL TPP resulted in the best spinach growth, while the treatment at 40°C, pH 6, and a TPP concentration of 20 mg/mL hindered growth; and the growth ration increased over 47.4% after the harvest than the normal growing spinach, the final biomass reached 2.53 g per plant. In addition, phenanthrene (PHE) and pyrene (PYR) predominantly accumulated more in the spinach roots, with variations depending on the specific CCNP treatment. And the exogenous application of CCNPs can reduce the PAH transfer to the shoots. The bioconcentration factors and transfer factors of PYR and PHE reduced differential movement within the spinach plants, and the spinach prefers PYR than PHE in the biological accumulation. This study offers a new understand that the mechanisms underlying NPs-PAHs interactions and the NPs implications for crop protection and food safety.
Q2. In the Introduction Section the Authors show the most important information on the nanoparticles. However, it does not include all relevant references. Indeed, more recently published papers should be cited.
Answer: Thanks for your good suggestion to improve our manuscript quality. It is important to show the importance of the NPs in the agriculture in the introduction. And we revised the importance cases of the NPs below,
“It was found that chitosan-coated mesoporous silica nanoparticles at the seedling stage led to a 70% increase in the fruit yield of uninfected watermelon because of their high surface area (Buchman et al., (2019). And a carrier system for paraquat using polymeric nanoparticles composed of chitosan/TPP can make paraquat less toxic and used for safer control of weeds in agriculture due to its controlled release (Grillo et al., (2014).” From line 91 to 95.
Reference
- T. Buchman, W. H. Elmer, C. Ma, K. M. Landy, J. C. White, C. L. Haynes, Chitosan-Coated Mesoporous Silica Nanoparticle Treatment of Citrullus lanatus (Watermelon): Enhanced Fungal Disease Suppression and Modulated Expression of Stress-Related Genes, ACS Sustainable Chemistry & Engineering, 2019, 7(24).
- Grillo, A. E. S. Pereira, C. S. Nishisaka, R. De Lima, K. Oehlke, R. Greiner, L. Fraceto, Chitosan/tripolyphosphate nanoparticles loaded with paraquat herbicide: An environmentally safer alternative for weed control, Journal of Hazardous Materials, 2014, 278:163-171.
Q3. The manuscript is prepared carelessly. For instance, the quality of the equations 1 and 2 should be improved.
Answer: Thanks for your good suggestion to improve our manuscript quality. We revised the quality of the figures and equations 1 and 2,
BCF = (1)
and
TF = (2)
Q4. The obtained results should be compared with those presented in the literature.
Answer: Thanks for your good suggestion to improve our manuscript quality. It is important to have a comparison with the literature to highlight our research results. We revised our conclusion part, and compared our results with the previous study.
Q5. The Conclusion Section is too short.
Answer: Thanks for your good suggestion to improve our manuscript quality. We revised conclusion in the manuscript and below,
These findings demonstrate that the reduction of PAHs transfer and accumulation within spinach plants by CCNPs is influenced by various factors, including the physicochemical properties of the nanoparticles and the specific exposure conditions. Particularly, CCNPs exhibit better performance at room temperature and neutral pH, making them more suitable as carriers for β-carotene in plants compared to conditions of high temperature and acidic pH. Under appropriate reaction conditions, the synthesized CCNPs show significant protective effects on plants by efficiently providing and releasing β-carotene in PAHs-contaminated environments. The decreased uptake and translocation of PAHs within plants can have significant implications for food safety and environmental health.
Furthermore, the main materials used in CCNPs synthesis are derived from natural and biologically harmless sources, indicating minimal risk in consuming them. Additionally, their sustainability is enhanced by the abundance of the precursor materials used in their synthesis. However, further research is necessary to fully comprehend the mechanisms of transfer and potential risks associated with the interaction between CCNPs and PAHs in plant systems. This knowledge will aid in understanding the sustainable agricultural benefits of CCNPs and their potential applications in environmental remediation.

Reviewer 4 Report
This manuscript is needed to be improved
1. The introduction section is needed to be included more current reference related to cs nps.
2. The rationale of research must be depicted clearly in the last section of introduction.
3. Discussion section is needed to be improved.
4. In conculsion, the hypothesis of research must be explained in order to support the research
Author Response
Reviewer #4
General comments: This manuscript is needed to be improved
Answer: Thanks for your good suggestion to improve our manuscript quality. We carefully revised manuscript, and hope our revision can meet your critical requirements.
Q1. The introduction section is needed to be included more current reference related to cs nps.
Answer: Thanks for your good suggestion to improve our manuscript quality. We added the related reference with CS NPs in the revision,
“It was found that chitosan-coated mesoporous silica nanoparticles at the seedling stage led to a 70% increase in the fruit yield of uninfected watermelon because of their high surface area (Buchman et al., (2019). And a carrier system for paraquat using polymeric nanoparticles composed of chitosan/TPP can make paraquat less toxic and used for safer control of weeds in agriculture due to its controlled release (Grillo et al., (2014).” From line 88 to 93.
Reference
- T. Buchman, W. H. Elmer, C. Ma, K. M. Landy, J. C. White, C. L. Haynes, Chitosan-Coated Mesoporous Silica Nanoparticle Treatment of Citrullus lanatus (Watermelon): Enhanced Fungal Disease Suppression and Modulated Expression of Stress-Related Genes, ACS Sustainable Chemistry & Engineering, 2019, 7(24).
- Grillo, A. E. S. Pereira, C. S. Nishisaka, R. De Lima, K. Oehlke, R. Greiner, L. Fraceto, Chitosan/tripolyphosphate nanoparticles loaded with paraquat herbicide: An environmentally safer alternative for weed control, Journal of Hazardous Materials, 2014, 278:163-171.
Q2. The rationale of research must be depicted clearly in the last section of introduction.
Answer: Thanks very much for your good suggestion to improve our manuscript quality. We added the description of our rationale of research that “Therefore, with the best carrier and release performance of CNPs, β-carotene is expected to show its role in improving the PAHs resistance and protecting the crop growth under PAHs contaminated environment.” At the end of the Introduction.
Q3. Discussion section is needed to be improved.
Answer: We agreed with your comment of discussion section. We revised the discussion part from Line 265 to 348.
Q4. In conclusion, the hypothesis of research must be explained in order to support the research
Answer: We agreed with your suggestion that the hypothesis should be emphasized in the final conclusion. We revised the conclusion part in the manuscript,
These findings demonstrate that the reduction of PAHs transfer and accumulation within spinach plants by CCNPs is influenced by various factors, including the physicochemical properties of the nanoparticles and the specific exposure conditions. Particularly, CCNPs exhibit better performance at room temperature and neutral pH, making them more suitable as carriers for β-carotene in plants compared to conditions of high temperature and acidic pH. Under appropriate reaction conditions, the synthesized CCNPs show significant protective effects on plants by efficiently providing and releasing β-carotene in PAHs-contaminated environments. The decreased uptake and translocation of PAHs within plants can have significant implications for food safety and environmental health.
Furthermore, the main materials used in CCNPs synthesis are derived from natural and biologically harmless sources, indicating minimal risk in consuming them. Additionally, their sustainability is enhanced by the abundance of the precursor materials used in their synthesis. However, further research is necessary to fully comprehend the mechanisms of transfer and potential risks associated with the interaction between CCNPs and PAHs in plant systems. This knowledge will aid in understanding the sustainable agricultural benefits of CCNPs and their potential applications in environmental remediation.

Round 2
Reviewer 1 Report
I confirm the publicstion, agree with answers and changes.
Reviewer 2 Report
Accept in present form
Reviewer 4 Report
The revised manuscript can be considered for further process.